# Healthcare workers' perspectives on the availability and use of mobile health technologies for disease diagnosis and treatment support in the Ashanti Region of Ghana

Ernest Osei[1,2], Felix Apiribu[3]*, Jonathan Kissi[4], Lydia Sarpomaa Asante[1], Sabina Ampon-Wireko[1], Tivani P. Mashamba-Thompson[2,5]

1 Department of Public Health, School of Public Health and Allied Sciences, Catholic University of Ghana, Sunyani, Ghana, 2 Discipline of Public Health Medicine, School of Nursing and Public Health, University of KwaZulu-Natal, Durban, South Africa, 3 Department of Nursing, College of Health Sciences, Kwame Nkrumah University Science and Technology, Kumasi, Ghana, 4 Department of Health Information Management, School of Allied Health Sciences, University of Cape Coast, Cape Coast, Ghana, 5 School of Health Systems and Public Health, Faculty of Health Sciences, University of Pretoria, Pretoria, South Africa

* fapiribu@yahoo.com

## Abstract

### Introduction

Considering the usefulness of mobile health (mHealth) technologies in healthcare delivery in low- and middle-income countries, including Ghana; there is a need to explore healthcare professionals' perspectives on the availability and use of mHealth for disease screening and treatment of patients' conditions. The study's main aim is to explore healthcare professionals' perspectives regarding the availability and use of mHealth applications for disease screening and management at point-of-care in Ghana's Ashanti Region.

### Materials and methods

We conducted in-depth interviews with healthcare professionals who use mHealth applications daily between July and September 2020. A purposive sampling strategy was employed to select healthcare professionals who have been using mobile health application tools to support healthcare delivery. The researchers conducted 14 in-depth interviews with healthcare professionals on the availability and use of mHealth applications to support disease diagnosis and treatment of patients' conditions. Data were transcribed, coded, arranged, and analyzed to determine categories and themes.

### Results

The study results demonstrated that healthcare workers had positive perceptions towards mHealth applications. Healthcare professionals identified significant challenges concerning mHealth applications: the high cost of data; lack of education or limited awareness; poor

**Data Availability Statement:** All the data generated for this study are available on: Apiribu, Felix (2022):

Supplementary - Data.docx. figshare. Figure.
https://doi.org/10.6084/m9.figshare.19446830.v1

**Funding:** The authors received no specific funding for this work.

**Competing interests:** The authors have declared that no competing interests exist.

mobile networks; unstable internet connectivity; erratic power supply; and unavailability of logistics. Healthcare professionals identified the following prerequisite strategies to strengthen the use and scale-up of mHealth applications: stable internet connectivity; creating awareness; supplying logistics; reducing the cost of data; and developing local mobile apps.

## Conclusions

The study results revealed the availability of mHealth applications at the individual level for disease screening and treatment support of patients' conditions. The study also showed several significant challenges facing mHealth applications which need to be addressed to guarantee the successful implementation and scaling-up of mHealth activities at all levels of healthcare delivery. Hence, future research should incorporate healthcare professionals' perspectives to completely understand mHealth implementation and scaling-up challenges and measures to inform policy regulations.

## Introduction

Mobile health interventions are recognized as essential tools designed to help deliver better healthcare services in all settings but are currently more widely available and commonly used in highly-resourced settings [1, 2]. In low- and middle-income countries (LMICs), mobile health applications are emerging strategies for screening health conditions, medication adherence, treatment and management of diseases, and others [3]. Mobile health (mHealth) technologies are revolutionizing healthcare delivery by giving patients quick access to critical services, personal data tracking apps, and remote access to healthcare professionals in LMICs [4–6].

In this study, we have defined mobile health as the use of mobile devices, their various components, and other related technologies in healthcare delivery [7–9]. The digital landscape around the world, including sub-Saharan Africa [SSA], has had a profound impact since the outbreak of severe acute respiratory syndrome coronavirus 2 (SARs-CoV-2) [10]. The SARs-CoV-2 pandemic outbreak has underscored the importance of robust digital health to help improve access to quality healthcare delivery in SSA [10].

In SSA, mobile phone penetration has substantially risen to the challenge of increasing and improving access to healthcare delivery. The Global System for Mobile Communications 2020 report reveals that at the end of 2019, 417 million individuals subscribed to mobile phone services in SSA [10]. Mobile phones are penetrating substantially in LMICs, including Ghana; consequently, employing mobile phones and their applications in healthcare provision could come to hard-to-reach populations more than the usual healthcare delivery approach [11, 12]. It is expected that by the end of 2025, there will be over one billion subscribers of mobile connections in SSA [10]. However, in Ghana, the mobile connection penetration rate as of 2020 was about 55% and is expected to rise sharply in 2025 [10].

Improving the health status of a population requires frequent use and widespread accessibility of healthcare services [13]. Accessing healthcare services adequately is very important in achieving universal health coverage [14]. However, poor access and quality healthcare services are major concerns for many people in Ghana, especially hard-to-reach populations, due to limited resources [13, 15]. In addition, studies have shown that Ghana and other SSA countries face the dual burden of infectious and non-infectious diseases, weakening the already fragile

healthcare systems [13, 16]. Some of Ghana's challenges of poor access to healthcare services could be poor service delivery, insufficient human resources, inadequate information and communication technology (ICT) infrastructure, poor information systems, weak internet connectivity, inadequate health infrastructure, and others [17–20].

The emerging evidence demonstrates that the Government of Ghana (GoG) is expanding healthcare access to remote areas by constructing new health facilities, upgrading and training more health professionals, developing ICT infrastructure, and several others [17]. The government of Ghana has recognized the need to build more mobile network infrastructure, improve internet connectivity, digitalize, and strengthen the existing healthcare systems. This is evident with the launch of the e-Health strategy to improve digital health to meet Ghana's growing demand for high-quality healthcare services [17, 21].

Research shows that healthcare professionals across the globe have varied opinions and views on mHealth applications [22–25]. For instance, in some LMICs, studies showed that healthcare professionals considered mHealth technologies and applications very useful in screening diseases and managing, and treating patients' health conditions [22, 26]. Additionally, research illustrated that in SSA, healthcare professionals perceived mHealth programs to be convenient and comfortable to adopt to assist healthcare delivery [23, 27]. Although, some studies have demonstrated that health workers in Nigeria and Kenya have shown positive perceptions toward the introduction and integration of mHealth into healthcare delivery [24, 25]. However, in Saudi Arabia and Iran, some healthcare professionals exhibited poor attitudes and negative perceptions concerning mHealth technologies to screen existing diseases, disease surveillance, and managing and treating patients' health conditions [28, 29].

Evidence shows that during the peak of the SARs-CoV-2 pandemic, healthcare professionals used mHealth applications for contact tracing and monitored remotely suspected SARs-CoV-2 cases in Asia and Latin America [30, 31]. Evidence from LMIC shows that, in this era of SARS-CoV-2, mHealth applications have been utilized for screening, diagnosis, risk assessment, tracking of real-time transmissions, and others in all settings [32, 33]. The use of mHealth applications could reduce the spread of SARs-CoV-2 and other infectious diseases in overcrowded emergency rooms and improve patient care [34–36].

Considering the usefulness of mHealth technologies and applications in healthcare delivery in LMICs, including Ghana [9, 37]. There is a need to explore healthcare professionals' perspectives on the availability and use of mHealth for disease diagnostics and management at point-of-care [POC] in Ghana's Ashanti Region. The study's main aim is to explore healthcare professionals' perspectives on the availability and use of mHealth applications for disease diagnostics and management at POC in the Ashanti Region of Ghana. We expect that the results of this current study will benefit GoG, Ghana health service and its development partners, non-governmental organizations, and other relevant stakeholders in enhancing the provision of healthcare services in Ghana and different similar settings.

## Materials and methods

This qualitative study was carried out as a follow-up to a survey study of 100 randomly chosen primary healthcare facilities in the Ashanti region of Ghana [37].

### Study design and setting

A qualitative study that used an in-depth interview guide to conduct face-to-face interviews for our data collection was employed. The in-depth interviews were carried out with healthcare professionals within the Ashanti region of Ghana. The Ashanti region is referred to as the business center of Ghana. This region has a population of over four million, seventy thousand per

the 2010 population census, and it is expected to rise sharply due to its high growth rate and its large population size [38]. The region has many healthcare facilities, but most are highly under-resourced [39]. Access to healthcare services in this region remains a significant challenge because of the poor healthcare infrastructure, unavailability of skilled health professionals, and others [39, 40]. The Ashanti region is faced with many infectious and chronic diseases compared to the other areas of Ghana [41]. According to the Ghana AIDS Commission, it is the second-highest HIV prevalence in the entire country [42, 43]. Also, the Ashanti Region is bedeviled with a high malaria prevalence rate compared to the other regions of Ghana [44].

Primary healthcare clinics with the highest and lowest availability and use of mHealth for disease diagnosis and treatment support by healthcare workers were selected to participate in this qualitative study. Healthcare professionals who are using mHealth applications at the primary healthcare facilities in the Ashanti region of Ghana were recruited. We defined healthcare professionals as categories of people who have received the requisite training and skills and are authorized to give healthcare services at various recognized healthcare facilities. Throughout the interview sections, we collected data on the availability and use of mHealth applications for disease diagnosis and treatment support. In addition, we employed face-to-face, in-depth interviews with the participants as new information was revealed, discovered some hidden practices, and provided more valuable insights to broaden our understanding of mHealth applications.

## Study population

Eligible participants were healthcare workers who work and reside in the Ashanti region of Ghana. We enrolled key highly skilled healthcare professionals who might have used or are using mHealth applications to support the provision of quality healthcare. Healthcare workers interested in our study agreed to participate and provided their consent before the interview began. Participants who were not interested and did not provide consent were excluded from the study. The data collection was conducted in English as all the participants were comfortable with the language.

**Sample size and sampling technique.**   A purposive sampling strategy was employed to recruit highly skilled healthcare professionals who are using information technology tools including mHealth applications to support healthcare delivery in the Ashanti Region of Ghana. The fourteen (14) healthcare professionals who were used as participants in this study were due to saturation point during the face-to-face in-depth interviews.

## In-depth interviews

We conducted 14 in-depth interviews using a structured interview guide. We invited highly skilled healthcare professionals who use mHealth applications daily to support healthcare delivery as participants. This was done to ascertain their perspectives on the availability and use of mHealth technologies to support disease diagnosis and treatment of conditions.

## Data collection

Our qualitative data were collected from highly experienced healthcare professionals using face-to-face in-depth interviews with an interview guide. The interview guide was piloted with two healthcare professionals from our study site, who were excluded from the final sample. With the support of research assistants, the principal researcher conducted the interviews using the interview guide (S2 Table), which contained several open-ended questions. The interview guide assisted us in ensuring that all the essential aspects of mHealth applications for disease screening or diagnosis and treatment support were comprehensively covered during each interview

section. The face-to-face interviews were carried out between July 2020 and September 2020 at different healthcare professionals' workplaces. After that, the necessary changes were made to the interview guide. We conducted the face-to-face interviews in English until saturation was reached, where no new information was unfolding from the interviews [45].

## Data analysis

The audio-recorded interviews were comprehensively transcribed verbatim into a Microsoft word document. Following this, the resulting texts were analyzed by employing thematic analysis. To start the analysis, broad themes were extracted from the transcripts, and then coded themes were identified. In constructing the themes, we took into account statements of meaning that were present in most of the relevant data. To ensure the credibility of the findings, independent coders verified or confirmed the themes extracted from the data. The data analysis was done simultaneously with the data collection process. It allowed the researchers to gradually focus their interviews and observations, and to decide how to test the conclusions that were emerging. The transcripts of the interviews were then uploaded into the NVivo version 12 software package for analysis. A codebook based on the major themes of this study was developed. We transformed the major themes into tree nodes and free nodes. Based on the codebook, the authors independently coded the transcriptions and verified them. The identified emerging themes and sub-themes are discussed below with quotes from the respondents.

## Ethics statement

The study received ethical approval from the Biomedical Research Ethics Committee of the University of KwaZulu-Natal (approval number: BREC/00000202/2019), the Ethics Review Committee of Ghana Health Service (approval number: GHS-ERC006/11/19), and site permission from all our recruitment sites. All the participants in the study provided signed informed consent forms before the commencement of the data collection process. The in-depth face-to-face interviews were carried out at each participant's convenient space and time where their confidentiality could not be compromised.

## Results

The healthcare professionals' age ranged from 38–46, with work experience from 12 to 22 years. All the healthcare professionals were from district hospitals, health centers, and clinics. There were 8 male participants representing 57.14% whiles the female participants were 6 representing 42.86%. For the professional categories of the participants for this study, there were 7 Nurses representing 50%, 3 Disease Control Officers representing 21.42%, 2 Physician Assistants representing 14.28%, 1 Certified Anesthetic, and 1 Pharmacist representing 7.14% each. The levels of education of the participants in this study were 10 postgraduates representing 71.42% and 4 undergraduates representing 28.57%. The demographic characteristics of the healthcare professionals who participated in this study are presented in Table 1.

### Emerging themes

Six main themes were identified: availability of mHealth for disease diagnosis and treatment support; acceptance of mHealth applications; mobile health applications complementing healthcare delivery; required skills and training in using mHealth applications; challenges in using or implementing mHealth applications and strategies to strengthen the use of mHealth applications. The main themes and sub-themes are presented in Table 2.

**Table 1. Characteristics of study participants who provided their perspectives on the availability and use of mobile health applications for disease diagnosis and treatment support in the Ashanti Region of Ghana (N = 14).**

| Participants ID | Age | Gender | Level of education | Working experience | Professional category |
|---|---|---|---|---|---|
| 1. | 39 | Female | Postgraduate | 12 | Nurse |
| 2. | 46 | Female | Postgraduate | 22 | Nursing supervisor |
| 3. | 40 | Male | Postgraduate | 20 | Nurse |
| 4. | 43 | Male | Postgraduate | 21 | Pharmacist |
| 5. | 39 | Male | Undergraduate | 12 | Critical Nurse |
| 6. | 44 | Male | Postgraduate | 19 | Disease control officer |
| 7. | 43 | Male | Postgraduate | 20 | Physician Assistant |
| 8. | 42 | Female | Postgraduate | 21 | Physician Assistant |
| 9. | 38 | Male | Undergraduate | 13 | Community Nurse |
| 10. | 42 | Female | Postgraduate | 19 | Public health nurse |
| 11. | 40 | Female | Undergraduate | 19 | Disease control officer |
| 12. | 43 | Male | Postgraduate | 21 | Disease control officer |
| 13. | 41 | Male | Postgraduate | 19 | Certified Anesthetic |
| 14. | 39 | Female | Undergraduate | 14 | Community Nurse |

## Availability of mHealth devices for disease diagnosis and treatment support

Our results showed that mobile health devices such as simple mobile phones, smartphones, and others are readily available to healthcare professionals at the personal level. However, the findings also revealed that mHealth devices are generally unavailable at the facility level to be used to support healthcare services. This implies that healthcare professionals use their mobile phones to support diagnostics and treatment of conditions of their patients. For example, one participant stated as follows:

**Table 2. Themes and sub-themes.**

| Themes | Sub-themes |
|---|---|
| Availability of mHealth devices for disease diagnosis and treatment support | |
| Acceptance of mHealth applications | Perception of mHealth for disease screening and treatment support |
| | Readiness of the current healthcare system |
| Mobile health applications complementing healthcare delivery | |
| Required skills and training in using mHealth applications | |
| Challenges in using mHealth applications | High cost of data |
| | Lack of education or limited awareness |
| | Poor mobile network and unstable internet connectivity |
| | Unstable power supply |
| | Unavailability of logistics |
| Strategies to strengthen the use of mHealth applications | Educating and training healthcare workers |
| | Stable internet connectivity |
| | Awareness creation |
| | Provision of logistics |
| | Develop mobile apps locally |
| | Reducing the cost of data |

*"Mobile health devices are not available at the facility level but are mostly available to health professionals since almost all of us have our smartphones to support disease screening and treatment in our facility"* **(Participant 8, Physician Assistant).**

*"They are available at the personal level, although doctors have their treatment guidelines usually on their desks, most often you will see them searching for new and advance guidelines that are emerging to support them in giving better directions and plans towards the treatment of their patients"* **(Participant 2, Nursing supervisor).**

## Acceptance of mHealth applications

The findings revealed that healthcare professionals had accepted the use of mHealth applications to support healthcare delivery. Their acceptance is corroborated by their positive perceptions towards using mHealth for disease diagnosis or screening and treatment support and the readiness of the current healthcare systems to accommodate mHealth applications.

**Perceptions of mHealth for disease screening and treatment support.**   The results demonstrated that most of the participants perceived mHealth applications as helpful and supportive in disease screening and treatment procedures of cases. Most of them stated that mHealth applications are valuable as they help healthcare professionals to know more about their patient's conditions and learn all the new emerging management guidelines. Mobile health applications are handy and helpful because they are readily available and make referencing very quick instead of resorting to the opening of books, especially when needed urgently.

*"Health professionals should be encouraged to use these devices as they assist them in giving quality healthcare to their patients in terms of efficiency, accurate screening, and diagnoses of diseases, improve treatment procedures, and management of patients' conditions"* **(Participant 5, Nurse).**

*"It is beneficial. Sometimes, in the consulting room, you will see some medical doctors using their mobile phones to search for diagnoses of certain diseases. So, I think it is a good device. It helps us get a concrete diagnosis of conditions since we can gather quality information on cases. It also provides us with current information on routine diagnoses of emerging conditions"* **(Participant 11, Disease control officer).**

**Readiness of the current healthcare system.**   Most of our participants stated that health workers and healthcare systems are ready to accommodate mHealth applications to deliver quality healthcare services to their clients. Some healthcare professionals believe that the current healthcare systems are ready for mobile health applications since some of the facilities have even gone paperless whiles others are in the process of digitalizing their systems. This is an indication that the current healthcare system is ready to accommodate this new technology to support healthcare services. Others say that the level of tolerance and positive attitudes towards using mHealth devices for screening and treatment guidelines are increasing gradually in most healthcare facilities.

*"I will say that in my former place of work, they are willing to accept mHealth applications to treat chronic patients by giving them reminders, especially for HIV conditions. They see if you default, and once they have your mobile number, they quickly get in touch with you"* **(Participant 1, Nurse).**

*"Well, I think with the level that we have reached, and even at the moment, especially with this SARs-CoV-2 outbreak, people have come to embrace technology. Healthcare professionals are ready to welcome this technology; if it's well organized and adequately structured, its usage can be done quickly"* **(Participant 3, Nurse).**

## Mobile health applications complementing healthcare delivery

Most of the participants agreed that mHealth applications could complement healthcare delivery by making their work easier and simpler for tracking any conditions. In addition, most healthcare professionals contend that they could perform their work faster and more accurately with the advent of mHealth applications.

*"Most mobile phones have health apps that help to monitor patients' blood pressure, patients' oxygen saturation rate, patients' pulse rate, respiration rate, and the fundamental vitality of a patient are equipped in these mobile health devices. They help determine the ranges for healthy living and the extremes. So, in my opinion, mHealth is good; it allows you to monitor the normal range. When a patient falls out of the normal range, the device will indicate it; you can conduct your physical assessment to confirm or rule it out as a professional"* **(Participant 5, Nurse).**

*"It is supportive, lessens our workload, and helps us improve healthcare services and everything. It has improved our timeliness in meeting our clients and has made our work much more manageable. It is easy to use, affordable, available, and has a high accuracy rate"* **(Participant 6, Disease Control Officer).**

## Required skills and training in using mHealth applications

It was found that some healthcare professionals have not received any formal training on how to use mHealth devices and their applications to support healthcare delivery. Nonetheless, most of them could use such apps to help healthcare due to their fundamental knowledge of computing and technology. However, the results demonstrated that few healthcare professionals had been given the necessary skills and training in using mHealth applications to support quality healthcare delivery.

*"We do not have formal training on how to use such devices, but through our own acquired skills and efforts, we can search for treatment guidelines and other things using mobile phones. Mobile health technology is very useful"* **(Participant 2, Nursing supervisor).**

*"l will say that I have received formal training on the use of mobile apps and others because of my public health background, but most of my other colleagues have not been given any training on how to use mHealth devices. My formal training came in as a result of going back to school, where I was given this training on how to use mHealth applications for screening, treatment procedures of cases, and others"* **(Participant 10, Public health nurse).**

*"Some of us in this facility have received some formal training on the use of these mobile apps on the tablets given to us by the regional health authorities to support the treatment of TB and HIV cases. We were also taught how to use the apps for surveillance, health monitoring, and tracking of diseases"* **(Participant 12, Disease Control Officer).**

## Challenges in using or implementing mHealth applications

Healthcare professionals raised several concerns regarding the use of mHealth applications for disease screening and treatment procedures of conditions in their various facilities. They are high cost of data, poor mobile network and unstable internet connectivity, lack of education or limited awareness, unstable power supply, and unavailability of logistics.

**High cost of data.** The results revealed that the high cost of data makes the frequent use of mHealth applications a significant challenge for healthcare professionals and health managers. In addition, the study showed that their facilities do not provide them with free data, and as such, buying data for mHealth applications has become very difficult for healthcare professionals.

*"The cost would be expensive to the healthcare workers and health managers. The cost of buying data will burden healthcare workers, and it will not encourage them to use mHealth. Buying data for an internet bundle will be costly for the health managers; for management, they are always not willing to do any little thing that brings cost"* **(Participant 1, Nurse).**

"In my opinion, the cost of data is expensive for health professionals and even sometimes to health managers, donor partners, and other development partners. For some of the mobile apps subscribing to them, you have to pay, and when they are in full use, you must pay as well. The cost involved in using some of these apps could discourage health authorities and users from considering using mHealth applications" **(Participant 5, Nurse).**

**Lack of education or limited awareness.** The results of our study showed that the majority of the participants have considerable knowledge of mHealth applications. However, a few healthcare professionals have limited knowledge of the use of mHealth applications for disease screening and treatment support. Most of them agreed that one of their significant challenges is the lack of education or unawareness of their patients and health managers.

*"I have first-hand experience with clients; sometimes, they misunderstand you when you hold mobile phones and think you don't care about them. Patients and their relatives are not aware that mHealth apps could deliver healthcare. For instance, I chanced on a nurse checking on a patient who came for hypoglycemia; we tried to check the respiratory distress syndrome in the baby, but we didn't have a glucometer that gave us the direct number, but we were using other methods to get the number. While I was trying to ask the nurse what she got, a relative nearly attacked me because I was using my phone to cross-check"* **(Participant 2, Nursing supervisor).**

**Poor mobile network coverage and unstable internet connectivity.** Our findings found that most healthcare professionals do not have access to internet connectivity. Even those who have access experienced unstable internet connectivity due to poor mobile network coverage.

*Sometimes we have weak mobile network coverage and even poor internet connectivity.* **(Participant 7, Physician Assistant).**

*Internet reliability is not strong enough in this facility (***Participant 5, Nurse).**

**Unstable power supply.** The findings from the study illustrated that most healthcare professionals do not have a stable power supply to help them fully utilize mHealth applications. They contend that several health facilities lack a stable power supply. They do not even have standby generators to support their work when taken off the national grid. This makes the continuous use of mHealth apps quite challenging for many healthcare professionals.

**Unavailability of logistics.**   Our results demonstrated that the unavailability of mHealth devices at the facility level is a significant challenge for healthcare professionals. Some raised concerns that health facilities should provide mobile health devices with recognized apps so they could use them at work instead of depending on individual ones. Others argued that there is no supply of free data for internet connectivity to assist them in getting new emerging diagnostic and treatment guidelines.

*"You know you have to use modern smartphones for internet connectivity, so when you don't have these smartphones, you cannot access the internet for current information on screening of cases and treatment guidelines"* **(Participant 8, Physician Assistant).**

## Strategies to strengthen the use of mHealth applications

Healthcare professionals presented their perspectives on the strategies that can be adopted to enhance mHealth applications to support healthcare delivery. They include educating and training healthcare workers; providing stable internet connectivity; creating awareness; providing logistics; developing mobile apps locally; and reducing the cost of data.

**Educating and training healthcare workers.**   The study participants suggested that healthcare professionals should be given the necessary skills and training on how to use mHealth applications appropriately and accurately. Others stated that some healthcare professionals are unaware of these mHealth applications, although they may have them, and as such formal training is essential. Healthcare professionals also suggested that education be extended to health managers and the general public on mHealth applications.

*"We should have formal training and education on mHealth apps, sensitizing the public on mHealth apps because it is not everyone who knows how to use it and even those who know how to use it, are they using the correct procedures"* **(Participant 8, Physician Assistant).**

*"First, if they want to implement mHealth applications entirely in the whole facility, there should be public education to inform them that the mobile phones they use for communication, however, can be used to support healthcare delivery. So that when they see us using mobile phones, they will not get any wrong impression"* [**Participant 7, Physician Assistant**].

**Stable internet connectivity.**   Most healthcare professionals recommended that to utilize mHealth applications fully, health managers should provide stable internet services. Others also suggested that internet services should be made free for the clinical category of health workers.

*"There should be robust internet connectivity in this facility even though we are managing with buying airtime and other stuff for now regarding communication using these mobile devices. We have an arrangement with Vodafone Ghana, so they have given us some phones that can be used for internal communication, but we have poor internet connectivity"* **(Participant 4, Pharmacist).**

*"One funny thing is that most experts are concentrated in urban areas where all these things are available, but when you go to a smaller facility where they don't have the needed human resources, they have to rely on mHealth applications. So, working in such facilities without internet connectivity becomes very challenging" (***Participant 7, Physician Assistant).**

**Awareness creation.**   Most participants recommended that creating awareness of mHealth applications for disease diagnosis and treatment procedures of patients' conditions is essential.

Some suggested strategies such as organizing workshops for healthcare professionals and health managers, community health education campaigns, radio and television advertisements, and others for the public.

**Provision of logistics.** Our study participants recommended that healthcare facilities provide each unit with these mobile health devices rather than depending on individual ones that could lead to abuse. In addition, they suggested that health managers collaborate with donor partners to procure mobile health devices and provide free data for internet connectivity, and other supporting devices. This will enhance the smooth implementation of mHealth applications in most health facilities.

*"At least if the health managers or any other donor partners can provide health workers with mobile phones for treatment and screening of diseases for each unit in our facility, it will help us a lot. They are providing free WIFI to the healthcare professionals to encourage the continued use of mHealth apps to support healthcare services"* **(Participant 11, Disease Control Officer).**

*"Procuring mHealth devices and buying data for each unit in this facility is better. We cannot compare the use of folders to these electronic devices. It is better and safer to use mHealth apps than the folders"* **(Participant 2, Nursing supervisor).**

**Develop mobile apps locally.** Our study participants revealed that localized mobile apps should be developed to suit our settings. They contend that most developed apps are designed for different environments, so their usage in this setting becomes very challenging for them.

*"If the standard treatment guidelines are well structured and developed locally in an app form so that we do not always have to rely on information from other places. This will make referencing quick and easy for healthcare providers to provide quality healthcare services. Also, a lot of apps could be developed for better interaction among us as health professionals to help us get answers to certain questions from a far distant area, so you do not have to go to Google for some information that sometimes becomes difficult to verify"* **(Participant 4, Pharmacist).**

**Reducing the cost of data.** Most healthcare professionals raised concerns about the high cost of data regarding the use of mHealth applications for disease screening and treatment of conditions. Therefore, in ensuring the effective implementation and scaling-up of mHealth applications by healthcare professionals, strategic measures should be put in place to subsidize the cost of data. In other words, healthcare professionals should be provided with either free WIFI or highly affordable data for internet connectivity. This will help them get current information on the new emerging diagnostic procedures and treatment guidelines.

*"If we can get a free WIFI in our settings, in a way it will reduce the cost that we incur as health professionals to be able to access health information to assist our clients in this facility"* **(Participant 13, Certified Anaesthetics).**

## Discussion

To the best of our knowledge, this present qualitative study is the first extensive study that focused on healthcare workers' views on the availability and use of mHealth applications for disease screening and treating patients' conditions in this region. Several key findings from this study deserve attention.

Our findings correspond to findings from a study conducted in Kenya that demonstrated that healthcare workers were highly comfortable and delighted with using mHealth applications to provide healthcare services [25]. The two studies revealed that healthcare professionals were enthusiastic about this technology because they have mHealth apps on their mobile phones with current emerging diagnostic and treatment guidelines which they could easily rely on when in need. Our current study shows that some healthcare workers in the Ashanti Region of Ghana have inadequate skills and training in using mHealth apps for disease screening and treating conditions. These findings have been shown in other resource-limited settings where some healthcare workers have insufficient training and skills to use mHealth applications to promote healthcare access [46–48]. In these studies, despite the insufficient training and skills, healthcare professionals intimated that mHealth applications such as text messages and mHealth apps are very useful.

Our study presented the current challenges experienced by healthcare professionals in using mHealth for disease screening and treating conditions in this region. These could inform and assist policymakers when developing regulations on mHealth applications in healthcare provision. Despite all the strengths, some limitations observed were: the study was carried out in the Ashanti region of Ghana for healthcare professionals' perspectives on mHealth applications for disease screening and treatment support; hence, the results cannot be generalized. In addition, the principal researcher carried out the interviews, which could introduce some form of bias. However, the interview guide was used to direct our conversation.

The study revealed several challenges to using mHealth applications for disease screening and management from healthcare workers' perspectives. First, policymakers need to adopt measures to support the smooth implementation and utilization of mHealth applications for disease diagnostics or screening and management at POC. Because of this, a proposed framework needed to improve the smooth implementation and utilization of mHealth applications for disease screening and treatment support in LMICs is recommended [12]. Second, the findings further indicated that healthcare workers had accepted mHealth applications for disease screening, treatment, and management of conditions. This is very encouraging as the continued utilization of mHealth applications at POC could help achieve universal coverage.

The results identified some significant challenges such as the high cost of data, limited awareness, poor mobile network coverage, unstable internet connectivity, and others affecting the utilization of mHealth applications. We, therefore, encourage health managers and other relevant stakeholders to help address these challenges to promote the use of mHealth applications. The findings illustrated the positive perceptions and the high level of tolerance towards mHealth applications by relevant stakeholders in healthcare service. We recommend that healthcare professionals continue to sensitize all stakeholders involved in healthcare provision to create more awareness to sustain the use of mHealth applications in various health facilities.

## Conclusion

The study results revealed the availability of mHealth applications at the individual level for disease screening and treatment support of patients' conditions. Therefore, we encourage policymakers to ensure that systems are put in place to check the use of mHealth applications at the individual level. The present study findings illustrated the significant challenges that faced healthcare professionals in using mHealth applications for disease screening and management at POC in resource-limited settings. Therefore, the researchers encourage collaborative efforts from policymakers, implementers, development partners, and other relevant healthcare stakeholders to address the challenges raised by healthcare professionals regarding mHealth applications.

## Supporting information

**S1 Table. Code book of data analysis conducted in NVivo software.**
(DOCX)

**S2 Table. Interview guide.**
(DOCX)

## Acknowledgments

The authors are grateful to the University of KwaZulu-Natal for providing us with essential research resources during this study. We thanked the 14 staff members at the primary health-care facilities who participated in this study. The authors would like to thank the Authorities of the Ashanti Regional Health Directorate, the District Health Management Teams, and all the PHCs managers for permitting us to conduct this study. Finally, we are grateful to the Department of Public Health Medicine staff for their support in diverse ways.

## Author Contributions

**Conceptualization:** Ernest Osei, Tivani P. Mashamba-Thompson.

**Data curation:** Ernest Osei, Felix Apiribu, Jonathan Kissi, Lydia Sarpomaa Asante, Sabina Ampon-Wireko, Tivani P. Mashamba-Thompson.

**Formal analysis:** Ernest Osei, Felix Apiribu, Jonathan Kissi, Lydia Sarpomaa Asante, Sabina Ampon-Wireko, Tivani P. Mashamba-Thompson.

**Investigation:** Ernest Osei.

**Methodology:** Ernest Osei, Tivani P. Mashamba-Thompson.

**Project administration:** Ernest Osei, Tivani P. Mashamba-Thompson.

**Supervision:** Tivani P. Mashamba-Thompson.

**Validation:** Jonathan Kissi, Lydia Sarpomaa Asante, Tivani P. Mashamba-Thompson.

**Writing – original draft:** Ernest Osei, Tivani P. Mashamba-Thompson.

**Writing – review & editing:** Ernest Osei, Felix Apiribu, Jonathan Kissi, Lydia Sarpomaa Asante, Sabina Ampon-Wireko, Tivani P. Mashamba-Thompson.

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
