## [Decision Letter · Decision Letter 0]

7 Jan 2022

PONE-D-21-22485Healthcare workers’ perspectives on the availability and use of mobile health technologies for disease diagnosis and treatment support in the Ashanti Region of GhanaPLOS ONE

Dear Dr. Apiribu,

Thank you for submitting your manuscript to PLOS ONE. After careful consideration, we feel that it has merit but does not fully meet PLOS ONE’s publication criteria as it currently stands. Therefore, we invite you to submit a revised version of the manuscript that addresses the points raised during the review process.

Two reviewers have provided extensive comments on how to improve the manuscript.

We look forward to receiving your revised manuscript.

Kind regards,

Nancy Beam, PhD

Staff Editor

PLOS ONE

Journal Requirements:

2. Please include your tables as part of your main manuscript and remove the individual files. Please note that supplementary tables (should remain/ be uploaded) as separate "supporting information" files".

Reviewers' comments:

Reviewer's Responses to Questions

**Comments to the Author**

1. Is the manuscript technically sound, and do the data support the conclusions?

Reviewer #1: Partly

Reviewer #2: Partly

2. Has the statistical analysis been performed appropriately and rigorously? 

Reviewer #1: N/A

Reviewer #2: N/A

3. Have the authors made all data underlying the findings in their manuscript fully available?

Reviewer #1: Yes

Reviewer #2: No

4. Is the manuscript presented in an intelligible fashion and written in standard English?

Reviewer #1: Yes

Reviewer #2: No

5. Review Comments to the Author

Reviewer #1: This paper reports on 14 interviews with health workers on mhealth. It is generally clearly written, and all appropriate steps for qualitative analysis have been followed.

Two sets of comments, the first a little more substantive, the second more minor edits.

More inter-related substantive comments

- As it stands, the paper feels rather thin, based as it is on high level opinions on all mhealth applications. But mhealth applications and the way in which they might be applied across all potential health-related decisions and actions are hugely variable (eg accessing the internet for information, follow-up reminders for different types of disease/care etc etc) raising questions about the real value of reporting such generic viewpoints (and in the discussion of comparing them to wider literature). I note a reference to a survey in reference 42. A suggestion would be to weave these findings if possible into the background, methods and findings (obviously ensuring proper reference to that published paper), in order to provide better context to qualitative views being shared.

- There is not much evidence of critical analysis of what's being said by participants - eg re accessing info/guidance on the internet (what's being looked up and is that considered appropriate), re training (what kinds of training and is that really valuable etc). Is what the health workers are discussing hypothetical (if they had the opportunity), or in practice (based on experience?). Possibly as above, linking the findings to the survey work would assist with interpreting/analysing the findings a little more for the reader.

There are some editorial issues the authors should correct but generally the English s strong and clear. The authors should really carefully review all the sentences and wording. For example first sentence in the abstract is incomplete, an error repeated again the main paper. Also, the first sentence of intro needs clarifying - instead of 'are used', do the authors mean 'are currently more widely available and commonly used in...'? And there are two sentences on page 5 second para that start... Although (the first sentence) followed by However (the second sentence). This is not an ideal flow. Also SARs - all words start with a capital or none?

Reviewer #2: I appreciate the effort made by the authors to describe their study. There are some worthwhile points made in the Results section, but overall, the manuscript has many language errors and text that is not understandable. It needs major re-organizing and re-writing to be clear, consistent, and useful to readers. I have added a number of specific comments and suggestions below to be helpful as the authors consider revisions.

Abstract

1. The Introduction of the Abstract says the study’s main aim is to identify challenges healthcare professionals face regarding the availability and use of mHealth applications for disease screening and management at point-of-care. The same point is made again on page 5, lines 132-134. The title suggests something more open. It’s important that the title and description of the study aim be consistent. Can the authors revise one of the other – title or description of the study aim?

2. The Methods section of the Abstract says “the data were categorized

into groups with the assistance of NVivo version 12. Our data was organized and

condensed into minor forms before coding began….” It is not clear what “categorized into groups” means. I know space is tight in the Abstract, so perhaps take this out and then make sure it is clarified in the main body of the manuscript.

Introduction

3. The first paragraph has a lot of useful information in it. It’s quite long though, so I would suggest breaking it up in the middle (roughly) to improve readability. A good place to start the next paragraph would be at line 86 “The digital landscape around the world including sub-Saharan Africa…”

4. On page 4, the paragraph beginning with “The literature revealed that healthcare professional used …” is hard to follow. It starts with how healthcare workers have used mHealth applications during the present COVID-19 era, and then goes into more general evidence on the views of healthcare workers toward mHealth tools (not related to COVID). It would be easier to follow the authors if use during COVID was described separately from the views of healthcare workers.

5. The authors write on page 4, in the paragraph beginning with “Considering the critical role that mHealth technologies…” that mHealth plays a “critical role” in promoting better healthcare services in SSA. However, they have not presented evidence to show this. They note that mHealth interventions are viewed as essential tools and that they represent emerging strategies for a variety of activities. They note that use of mobile phones is rising. They describe a variety of challenges providing care in Ghana and other settings, but also note that Ghana needs to build more mobile network infrastructure. They then describe the views of healthcare workers towards mHealth approaches. I do not see anything demonstrating that mHealth strategies actually have improved healthcare services in SSA. Can the authors please provide such evidence or revise this statement accordingly?

Materials and Methods

6. On page 6, lines 143-44: what does “interpretive paradigm” mean? Is the next sentence meant to clarify what it means? I’m afraid it wasn’t clear to me and I found the paragraph confusing.

7. The first two paragraphs in this section have overlapping and duplicative information. I would suggest putting the information describing the Ashanti region that is now in the second paragraph into the first paragraph, right after the first sentence that describes the main design. Then the authors could move all the information about the IDIs themselves into the second paragraph. This would improve the flow.

8. There seems to be conflicting information about who was eligible. In the first paragraph under “Study population” (page 7, lines 174-176) the authors say that “All the potential participants were invited to participate in this qualitative study…” However, the next paragraph starts by saying that the research team purposively selected 14 healthcare professionals…” (lines 181-182). Can the authors please clarify how recruitment took place?

9. The authors write in the middle of the paragraph under “Sampling and sample size” that the study was carried out as a follow up to survey (lines 185-187). This is important framing of the present study. It should go at the beginning of the Materials and Methods section and not be buried in this paragraph.

10. The last sentence in the paragraph under “Sampling and sample size” on selection of the primary healthcare clinics where the study took place should go before a description of the healthcare workers to help guide the reader and help them appreciate the context of this work.

11. Page 8, line 206: what does “We condensed the data into more minor forms” mean?

12. The Ethics section on page 8 is a bit confusing. Could the authors confirm that every participant provided signed informed consent before participating in the study?

Results

13. In the paragraph under “Emerging themes” the authors write that six themes were identified, but only five themes seem to be described here. Please revise so that the language is consistent. (It seems from reading further into the Results that the theme “Challenges in using or implementing mHealth applications” (page 12, line 340) appears to have been left out of the description of the six main themes.)

14. Out of respect for participants, I believe it is important that the direct statements of participants be presented without major grammatical errors. Sometimes such errors are carried over into a manuscript or report due to imperfect translation from another language into English, but here the authors say that the participants were comfortable using English. I would recommend that all direct statements be corrected so that they are grammatically correct and understandable. An example of a statement that would benefit from some slight revisions is the one on lines 248-249 on page 9: “Yes, they are mostly available because personally, most health staff have their phones, but unavailable at the facility level, in Ghana which facility will purchase mobile phones for use.”

Discussion

15. The first two paragraphs of the discussion summarize the main findings in quite a bit of detail. I think that’s not necessary. Rather, is there a way to indicate here why the study was unique or novel, or contributes something new? Perhaps this is the first such study in Ghana?

16. The third paragraph of the discussion seems to trying to connect the study findings to other research, which is really good. However, it covers a number of points and goes on for a long time. I would suggest breaking it up and focusing on only one point in the results and the associated research. This will help the flow for the reader enormously.

17. The paragraph beginning on line 518 (the in-depth interviews created an opportunity…”) contain information on the strengths of the study. While I don’t believe all the points here are necessary, I would suggest moving several points here to the first paragraph of the discussion section as they help illustrate why the study is important.

Language issues

18. There are many grammatical errors and confusing sentences or text in the manuscript. An example is the first sentence of the paragraph beginning on line 149 on page 6. The sentence is not a complete sentence. Another example is the sentence on page 8, lines 213-214, beginning “This was done simultaneously…” This sentence is very confusing – I do not understand what it means.

6. PLOS authors have the option to publish the peer review history of their article (what does this mean?). If published, this will include your full peer review and any attached files.

Reviewer #1: No

Reviewer #2: No

---

## [Author Response · Author response to Decision Letter 0]

29 Mar 2022

A letter responding to the reviewers comments have been uploaded for your attention. kind regards

---

## [Decision Letter · Decision Letter 1]

10 Feb 2023

PONE-D-21-22485R1Healthcare workers’ perspectives on the availability and use of mobile health technologies for disease diagnosis and treatment support in the Ashanti Region of GhanaPLOS ONE

Dear Dr. Apiribu,

Thank you for submitting your manuscript to PLOS ONE. After careful consideration, we feel that it has merit but does not fully meet PLOS ONE’s publication criteria as it currently stands. Therefore, we invite you to submit a revised version of the manuscript that addresses the points raised during the review process.

We look forward to receiving your revised manuscript.

Kind regards,

Philip Teg-Nefaah Tabong

Academic Editor

PLOS ONE

Reviewers' comments:

Reviewer's Responses to Questions

**Comments to the Author**

1. If the authors have adequately addressed your comments raised in a previous round of review and you feel that this manuscript is now acceptable for publication, you may indicate that here to bypass the “Comments to the Author” section, enter your conflict of interest statement in the “Confidential to Editor” section, and submit your "Accept" recommendation.

Reviewer #2: (No Response)

2. Is the manuscript technically sound, and do the data support the conclusions?

Reviewer #2: Partly

3. Has the statistical analysis been performed appropriately and rigorously? 

Reviewer #2: Yes

4. Have the authors made all data underlying the findings in their manuscript fully available?

Reviewer #2: No

5. Is the manuscript presented in an intelligible fashion and written in standard English?

Reviewer #2: No

6. Review Comments to the Author

Reviewer #2: It was challenging to review revisions in the absence of a letter with replies to the reviewers’ comments. I can see some changes were made to the manuscript, but there are still some issues, including some raised earlier that were not addressed in this revised version, as best I can tell. Below are some more detailed comments that I hope are helpful.

Abstract

1. I can see that the text describing the study’s main aim has been changed to better align with the title. This is helpful.

The Methods section of the first Abstract we see in the pdf says “the data were categorized into groups with the assistance of NVivo version 12. Our data was organized and condensed into minor forms before coding began….” I commented earlier that it is not clear what “categorized into groups” means and suggested either explaining it, or just taking it out of the Abstract if the word limit was too tight. It looks like no change here was made here, but then in the Abstract on page 2 of the pdf, it looks like that part of the sentence was removed. The text on page 2, lines 35-37, says “Data were organized and broken down into smaller units before being coded” is also unclear. How were the data “broken down” and what does “smaller units” mean?

Introduction

2. A minor point, but it is hard to read the introduction given the lack of clear paragraphs. It seems the authors tried to break up previously very long paragraphs (a good thing), but the way the text reads now, without clear breaks, it remains a bit hard to follow.

3. On page 4, the paragraph beginning with “The literature revealed that healthcare professional used …” is hard to follow. It starts with how healthcare workers have used mHealth applications during the present COVID-19 era, and then goes into more general evidence on the views of healthcare workers toward mHealth tools (not related to COVID). It would be easier to follow the authors if use during COVID was described separately from the views of healthcare workers.

Materials and Methods

4. On page 5, lines 126-128, the authors provide some explanation of what “interpretive paradigm” means, in response to an earlier question. I appreciate the revised text, but it’s just not clear what the “interpretive paradigm” is. Can the authors provide a definition? This will help readers understand what they are doing in this manuscript.

5. On page 6, lines 156-169, the text is still confusing about how the study team selected the 14 healthcare professionals. Lines 166-168 suggest it was really a matter of choosing representative healthcare clinics? If that is the case, can the authors describe the kinds of things they were trying to balance among the healthcare clinics—for instance, size, location?

6. Page 7, line 183, the paragraph beginning with “We employed content analysis method to analyze our data…” is hard to understand. What does “The data was broken down into smaller units” mean? This question is also relevant for the description of analysis in the abstract. It’s just not clear how this analysis was done.

7. The data management section, beginning on page 7, line 189, would work better before the description of data analysis (now beginning on line 183).

Results

8. In some sections of the Results, the text is very brief and light on content, while there are numerous direct quotes. It seems as if the quotes contain the results rather than illustrating the results (as examples). An example of this is in the section headed “Readiness of the current healthcare system” beginning on page 12, line 269, where there is one sentence followed by 4 direct quotes of participants. It would strengthen the Results section if the authors provide more of their own analysis by fleshing out some of the main points in their own words. If space is tight, one or two direct quotes in each section could be cut.

Discussion

9. The first paragraph now starts by saying that this is the first extensive study on availability and use of mHealth applications in the Ghana region, which is a strong start to the discussion. However, the next few sentences restate aspects of the methods to stress how they were appropriate and relevant for the study. Some of these points seem like a stretch and I’d recommend cutting them. An example is: “We also used note-taking and transcript validations of participants to boost the credibility of our study findings.” I don’t see any previous mention of “transcript validations” and am not entire clear on what this means – I would cut this.

10. The second paragraph of the discussion repeats key findings, which isn’t really necessary. I would recommend ending the first paragraph with “Several key findings deserve attention.” (or something like that) And then start the next paragraph with the third paragraph. This means cutting the 2nd paragraph.

11. The third paragraph notes some of the authors’ key findings, and notes alignment with other research. This is just what the discussion should do, but nothing very interesting about the alignment/agreement is noted. For instance, in the first point, in lines 481-484, could the authors add something about WHY healthcare workers in both studies said they were comfortable using mHealth applications – did participants in both studies say, for instance, that they felt text messages were an effective communication tool with patients, or that they thought it was fun to use this technology? In the second point, the authors say that their finding that “some healthcare workers had limited awareness of mHealth applications….” agreed with a study in South Africa. This point begs the question – how does this finding contradict (or not) the first point about participants being enthusiastic about use of mHealth technology? This seems worthy of a bit more discussion—what do the authors think about these two findings (which on the face of it, seem contradictory)? The goal of the discussion is to flesh out the findings and say how they add to our understanding of the main study question. Currently, the discussion could be much stronger in this regard.

12. The next paragraph beginning with “The in-depth interviews created an opportunity…” is still a list of study aspects that just feels inappropriate. Study strengths are now described at the beginning of the discussion. Most of the points in this paragraph are not unusual or add anything. I would cut it. The next three paragraphs have good content and stand on their own well.

Language issues

13. The language is better than before (nice job with editing!) but there are still many grammatical errors in the manuscript. Examples are: page 3, line 69: the sentence beginning with “Global System for …” should start with “The” and page 4, line 86, “Emerging evidence…” isn’t correct. It could be “The emerging evidence…” or “The evidence to date…”

7. PLOS authors have the option to publish the peer review history of their article (what does this mean?). If published, this will include your full peer review and any attached files.

Reviewer #2: No

---

## [Author Response · Author response to Decision Letter 1]

4 Mar 2023

We are most grateful for reviewing our manuscript titled: Healthcare workers’ perspectives on the availability and use of mobile health technologies for disease diagnosis and treatment support in the Ashanti Region of Ghana and providing constructive feedback. We have considered your comments and recommendations. 

Please find our responses to your comments, queries, and suggestions. All revisions have been highlighted in yellow in our main manuscript. Any further comments will be addressed and modified by us.

Reviewer #2: Comments

It was challenging to review revisions in the absence of a letter with replies to the reviewers’ comments. I can see some changes were made to the manuscript, but there are still some issues, including some raised earlier that were not addressed in this revised version, as best I can tell. Below are some more detailed comments that I hope are helpful. 

 Abstract

1. I can see that the text describing the study’s main aim has been changed to better align with the title. This is helpful.

The Methods section of the first Abstract we see in the pdf says “the data were categorized into groups with the assistance of NVivo version 12. Our data was organized and condensed into minor forms before coding began….” I commented earlier that it is not clear what “categorized into groups” means and suggested either explaining it, or just taking it out of the Abstract if the word limit was too tight. It looks like no change here was made here, but then in the Abstract on page 2 of the pdf, it looks like that part of the sentence was removed. The text on page 2, lines 35-37, says “Data were organized and broken down into smaller units before being coded” is also unclear. How were the data “broken down” and what does “smaller units” mean?

Thank you.

This has been revised to bring clarity. Please, refer to Materials and methods of Abstract, page 2, lines 36-37.

Introduction

2. A minor point, but it is hard to read the introduction given the lack of clear paragraphs. It seems the authors tried to break up previously very long paragraphs (a good thing), but the way the text reads now, without clear breaks, it remains a bit hard to follow.

This has been revised accordingly

3. On page 4, the paragraph beginning with “The literature revealed that healthcare professional used …” is hard to follow. It starts with how healthcare workers have used mHealth applications during the present COVID-19 era, and then goes into more general evidence on the views of healthcare workers toward mHealth tools (not related to COVID). It would be easier to follow the authors if use during COVID was described separately from the views of healthcare workers.

This has been amended accordingly. Please, refer to Background, page 4, lines 107-113.

Materials and Methods

4. On page 5, lines 126-128, the authors provide some explanation of what “interpretive paradigm” means, in response to an earlier question. I appreciate the revised text, but it’s just not clear what the “interpretive paradigm” is. Can the authors provide a definition? This will help readers understand what they are doing in this manuscript.

This whole section has been deleted completely 

5. On page 6, lines 156-169, the text is still confusing about how the study team selected the 14 healthcare professionals. Lines 166-168 suggest it was really a matter of choosing representative healthcare clinics? If that is the case, can the authors describe the kinds of things they were trying to balance among the healthcare clinics—for instance, size, location?

 This whole section has been amended accordingly. Please, refer to Materials and methods, page 6, lines 160-163.

6. Page 7, line 183, the paragraph beginning with “We employed content analysis method to analyze our data…” is hard to understand. What does “The data was broken down into smaller units” mean? This question is also relevant for the description of analysis in the abstract. It’s just not clear how this analysis was done.

 This has been amended accordingly. Please, refer to Materials and methods, Data analysis section, page 7, lines 178-190.

7. The data management section, beginning on page 7, line 189, would work better before the description of data analysis (now beginning on line 183).

 The data management section has been collapsed and fused into the Data analysis section, page 7, lines 178-190.

Results

8. In some sections of the Results, the text is very brief and light on content, while there are numerous direct quotes. It seems as if the quotes contain the results rather than illustrating the results (as examples). An example of this is in the section headed “Readiness of the current healthcare system” beginning on page 12, line 269, where there is one sentence followed by 4 direct quotes of participants. It would strengthen the Results section if the authors provide more of their own analysis by fleshing out some of the main points in their own words. If space is tight, one or two direct quotes in each section could be cut. 

This has been amended accordingly throughout the results section

Discussion

9. The first paragraph now starts by saying that this is the first extensive study on availability and use of mHealth applications in the Ghana region, which is a strong start to the discussion. However, the next few sentences restate aspects of the methods to stress how they were appropriate and relevant for the study. Some of these points seem like a stretch and I’d recommend cutting them. An example is: “We also used note-taking and transcript validations of participants to boost the credibility of our study findings.” I don’t see any previous mention of “transcript validations” and am not entire clear on what this means – I would cut this.

This has been deleted as recommended by the reviewers

10. The second paragraph of the discussion repeats key findings, which isn’t really necessary. I would recommend ending the first paragraph with “Several key findings deserve attention.” (or something like that) And then start the next paragraph with the third paragraph. This means cutting the 2nd paragraph.

This has been done as recommended by the reviewers

11. The third paragraph notes some of the authors’ key findings, and notes alignment with other research. This is just what the discussion should do, but nothing very interesting about the alignment/agreement is noted. For instance, in the first point, in lines 481-484, could the authors add something about WHY healthcare workers in both studies said they were comfortable using mHealth applications – did participants in both studies say, for instance, that they felt text messages were an effective communication tool with patients, or that they thought it was fun to use this technology? 

In the second point, the authors say that their finding that “some healthcare workers had limited awareness of mHealth applications….” agreed with a study in South Africa. This point begs the question – how does this finding contradict (or not) the first point about participants being enthusiastic about use of mHealth technology? This seems worthy of a bit more discussion—what do the authors think about these two findings (which on the face of it, seem contradictory)? The goal of the discussion is to flesh out the findings and say how they add to our understanding of the main study question. Currently, the discussion could be much stronger in this regard.

This has been amended according, please, refer to Discussion, page 18, lines 445-456.

It has been deleted to clear all forms of confusion

12. The next paragraph beginning with “The in-depth interviews created an opportunity…” is still a list of study aspects that just feels inappropriate. Study strengths are now described at the beginning of the discussion. Most of the points in this paragraph are not unusual or add anything. I would cut it. The next three paragraphs have good content and stand on their own well.

This has been removed as recommended by the reviewers

Language issues

13. The language is better than before (nice job with editing!) but there are still many grammatical errors in the manuscript. Examples are: page 3, line 69: the sentence beginning with “Global System for …” should start with “The” and page 4, line 86, “Emerging evidence…” isn’t correct. It could be “The emerging evidence…” or “The evidence to date…”

We are grateful for your observations.

Please, refer to the specific grammatical errors observed, and corrections made, Introduction section, page 3, lines 71-73, page 4, lines 89-91.

All grammatical errors in the entire manuscript have been corrected.

---

## [Decision Letter · Decision Letter 2]

1 Sep 2023

PONE-D-21-22485R2Healthcare workers’ perspectives on the availability and use of mobile health technologies for disease diagnosis and treatment support in the Ashanti Region of GhanaPLOS ONE

Dear Dr. Apiribu,

Thank you for submitting your manuscript to PLOS ONE. After careful consideration, we feel that it has merit but does not fully meet PLOS ONE’s publication criteria as it currently stands. Therefore, we invite you to submit a revised version of the manuscript that addresses the points raised during the review process.

We look forward to receiving your revised manuscript.

Kind regards,

Augustina Koduah

Academic Editor

PLOS ONE

Reviewers' comments:

Reviewer's Responses to Questions

**Comments to the Author**

1. If the authors have adequately addressed your comments raised in a previous round of review and you feel that this manuscript is now acceptable for publication, you may indicate that here to bypass the “Comments to the Author” section, enter your conflict of interest statement in the “Confidential to Editor” section, and submit your "Accept" recommendation.

Reviewer #3: (No Response)

Reviewer #4: (No Response)

2. Is the manuscript technically sound, and do the data support the conclusions?

Reviewer #3: Yes

Reviewer #4: Yes

3. Has the statistical analysis been performed appropriately and rigorously? 

Reviewer #3: Yes

Reviewer #4: Yes

4. Have the authors made all data underlying the findings in their manuscript fully available?

Reviewer #3: Yes

Reviewer #4: No

5. Is the manuscript presented in an intelligible fashion and written in standard English?

Reviewer #3: Yes

Reviewer #4: Yes

6. Review Comments to the Author

Reviewer #3: I would like to thank the authors for this interesting work.

mHealth in abstract: For the first time, it should be written in full, not abbreviated.

Similar works have been done in Iran, such as:

Ershad Sarabi, Roqhayeh, et al. "Role of mobile technology in Iran healthcare system: A review study." Journal of Health and Biomedical Informatics 4.4 (2018): 313-326.

Saeidnia HR, Mohammadzadeh Z, Hassanzadeh M. Evaluation of Mobile Phone Healthcare Applications During the Covid-19 Pandemic. Stud Health Technol Inform. 2021 May 27;281:1100-1101. doi: 10.3233/SHTI210363. PMID: 34042856.

Reviewer #4: Overall the manuscript is well-written and the revisions made, based on comments from the previous reviews, have improved it significantly. However, I have few comments that require the authors' attention before the manuscript is accepted for publication.

1. Methods Section

(i). There is lack of clarity on how the 14 healthcare professionals were recruited. On what basis were they recruited? How was the sample size of 14 determined? What sampling technique was adopted and the reasons for it? The authors should include a section on "Sample size and sampling technique" where these can be addressed.

(ii). In line 168, the authors write "With the support of the trained researchers, the principal researcher conducted the interviews

169 using the interview guide..." . However, there was no prior mention of any training for researchers. I suppose the authors meant to say trained "research assistants" instead of "the trained researchers" Please, clarify.

2. Results Section

Line 200-204: The authors summarised the age and work experience of participants in the first paragraph under Results. They could perform descriptive statistics to summarise the other demographic characteristics of the participants and include them in that paragraph. This would make it easier for readers to appreciate the characteristics of the participants and provide a general context to the results presented. Table 1 can remain as it is as it shows the background of the individuals who are quoted in the results section.

3. Language

The authors have attempted to improve the readability of the manuscript following the suggestions and comments from the previous reviews. However, there are still some grammatical errors in the manuscript. I recommend a professional language editing prior to acceptance. Some instances are highlighted below:

(i). Abstract: The first sentence in the introduction section is incomplete. It appears it can be complete by merging with the second sentence. This was pointed out in previous review but has still not been corrected.

(ii). Line 58: check the hyphen in "highly resourced-settings". It could be written as highly-resourced settings or "high-resource settings"; and in "low-and middle-income countries" there should be spacing between "low-" and "and" as in "low- and middle-income countries.

4. Data availability statement appears incomplete. The authors mention that data is available without indicating where or how the data can be accessed. Please refer to relevant journal policy/guidelines.

7. PLOS authors have the option to publish the peer review history of their article (what does this mean?). If published, this will include your full peer review and any attached files.

Reviewer #3: No

Reviewer #4: No

---

## [Author Response · Author response to Decision Letter 2]

1 Nov 2023

PLOS ONE Decision: Revision required [PONE-D-21-22485R2] - [EMID:62f8663b94d5c5b0]

PONE-D-21-22485R2

JOURNAL: PLOS ONE

Title: Healthcare workers’ perspectives on the availability and use of mobile health technologies for disease diagnosis and treatment support in the Ashanti Region of Ghana

Dear Reviewers 

We are most grateful for reviewing our manuscript titled: Healthcare workers’ perspectives on the availability and use of mobile health technologies for disease diagnosis and treatment support in the Ashanti Region of Ghana and providing constructive feedback. We have considered your comments and recommendations. 

Please find our responses to your comments, queries, and suggestions. All revisions have been highlighted in yellow in our main manuscript. Any further comments will be addressed and modified by us.

Reviewer #3: Comments

I would like to thank the authors for this interesting work.

mHealth in abstract: For the first time, it should be written in full, not abbreviated.

 We are most grateful.

This has been revised accordingly. Please, kindly refer to the Abstract section, lines 27-30.

Similar works have been done in Iran, such as:

Ershad Sarabi, Roqhayeh, et al. "Role of mobile technology in Iran healthcare system: A review study." Journal of Health and Biomedical Informatics 4.4 (2018): 313-326.

Saeidnia HR, Mohammadzadeh Z, Hassanzadeh M. Evaluation of Mobile Phone Healthcare Applications During the Covid-19 Pandemic. Stud Health Technol Inform. 2021 May 27;281:1100-1101. doi: 10.3233/SHTI210363. PMID: 34042856. We are grateful for suggesting these articles to us

Reviewer #4: Comments

Overall, the manuscript is well-written, and the revisions made, based on comments from the previous reviews, have improved it significantly. 

However, I have few comments that require the authors' attention before the manuscript is accepted for publication.

 We are grateful for your observations.

We have considered all the suggestions made by the Reviewer

1. Methods Section

(i). There is lack of clarity on how the 14 healthcare professionals were recruited. On what basis were they recruited? How was the sample size of 14 determined? What sampling technique was adopted and the reasons for it? The authors should include a section on "Sample size and sampling technique" where these can be addressed.

This has been revised accordingly as suggested by the reviewer. Please, kindly refer to Methodology section, page 6, lines 160 – 165. 

(ii). In line 168, the authors write "With the support of the trained researchers, the principal researcher conducted the interviews. 

169 using the interview guide..." . However, there was no prior mention of any training for researchers. I suppose the authors meant to say trained "research assistants" instead of "the trained researchers" Please, clarify.

This has been clarified. Please, kindly refer to Methodology section, page 6, lines 175 -176. 

2. Results Section

Line 200-204: The authors summarised the age and work experience of participants in the first paragraph under Results. They could perform descriptive statistics to summarise the other demographic characteristics of the participants and include them in that paragraph. This would make it easier for readers to appreciate the characteristics of the participants and provide a general context to the results presented.

Table 1 can remain as it is as it shows the background of the individuals who are quoted in the results section.

This has been revised accordingly as suggested by the reviewer. Please, kindly refer to the Result section, page 8, lines 209 – 216.

3. Language 

The authors have attempted to improve the readability of the manuscript following the suggestions and comments from the previous reviews. However, there are still some grammatical errors in the manuscript. I recommend a professional language editing prior to acceptance. Some instances are highlighted below:

We are grateful to the reviewer for this observation.

All grammatical errors have been addressed accordingly.

(i). Abstract: The first sentence in the introduction section is incomplete. It appears it can be complete by merging with the second sentence. This was pointed out in previous review but has still not been corrected. 

This has been amended by merging the first and second sentences in the abstract section, please, kindly refer to lines 27 – 30.

(ii). Line 58: check the hyphen in "highly resourced-settings". It could be written as highly-resourced settings or "high-resource settings"; and in "low-and middle-income countries" there should be spacing between "low-" and "and" as in "low- and middle-income countries.

This has been amended and revised accordingly, please, kindly refer to line 59.

4. Data availability statement appears incomplete. The authors mention that data is available without indicating where or how the data can be accessed. Please refer to relevant journal policy/guidelines.

This has been revised accordingly, please, kindly refer to lines 523 - 524.

---

## [Editor Report · Decision Letter 3]

6 Nov 2023

PONE-D-21-22485R3Healthcare workers’ perspectives on the availability and use of mobile health technologies for disease diagnosis and treatment support in the Ashanti Region of GhanaPLOS ONE

Dear Dr. Apiribu,

Thank you for submitting your manuscript to PLOS ONE. After careful consideration, we feel that it has merit but does not fully meet PLOS ONE’s publication criteria as it currently stands. Therefore, we invite you to submit a revised version of the manuscript that addresses the points raised during the review process.

Reviewer comments are adequately addressed. However, the portions on study participants under the results session (page 8, including the Table 1) is part of the method session. Authors should consider moving the portion on study participants to the method session.

We look forward to receiving your revised manuscript.

Kind regards,

Augustina Koduah

Academic Editor

PLOS ONE

Journal Requirements:

Additional Editor Comments:

Reviewer comments are adequately addressed. However, the portions on study participants under the results session (page 8, including the Table 1) is part of the method session. Authors should consider moving the portion on study participants to the method session.

---

## [Author Response · Author response to Decision Letter 3]

8 Nov 2023

PLOS ONE Decision: Revision required [PONE-D-21-22485R3]-[EMID: 279b1db1e8cfb81a] 

PONE-D-21-22485R3

JOURNAL: PLOS ONE

Title: Healthcare workers’ perspectives on the availability and use of mobile health technologies for disease diagnosis and treatment support in the Ashanti Region of Ghana

Dear Reviewers 

We are most grateful for reviewing our manuscript titled: Healthcare workers’ perspectives on the availability and use of mobile health technologies for disease diagnosis and treatment support in the Ashanti Region of Ghana and providing constructive feedback. We have considered your comments and recommendations. 

Please find our responses to your comments, queries, and suggestions. All revisions have been highlighted in yellow in our main manuscript. Any further comments will be addressed and modified by us.

Reviewer: Comments

Reviewer comments are adequately addressed.

However, the portions on study participants under the results session (page 8, including the Table 1) is part of the method session.

Authors should consider moving the portion on study participants to the method session We are very grateful to the Reviewer

Author response: 

We are very grateful to the Reviewer

Well noted

We have revised accordingly as suggested by the reviewer, kindly refer to lines 159 – 164, pages 6- 7.

---

## [Editor Report · Decision Letter 4]

10 Nov 2023

Healthcare workers’ perspectives on the availability and use of mobile health technologies for disease diagnosis and treatment support in the Ashanti Region of Ghana

PONE-D-21-22485R4

Dear Dr. Apiribu,

We’re pleased to inform you that your manuscript has been judged scientifically suitable for publication and will be formally accepted for publication once it meets all outstanding technical requirements.

Kind regards,

Augustina Koduah

Academic Editor

PLOS ONE
---

## [Editor Report · Acceptance letter]

3 Apr 2024

PONE-D-21-22485R4 

PLOS ONE

Dear Dr. Apiribu, 

I'm pleased to inform you that your manuscript has been deemed suitable for publication in PLOS ONE. Congratulations! Your manuscript is now being handed over to our production team.

Kind regards, 

on behalf of

Dr. Augustina Koduah 

Academic Editor

PLOS ONE